# Patterns, Management, and Outcome of Traumatic Femur Fracture: Exploring the Experience of the Only Level 1 Trauma Center in Qatar

**DOI:** 10.3390/ijerph18115916

**Published:** 2021-05-31

**Authors:** Syed Imran Ghouri, Mohammad Asim, Fuad Mustafa, Ahad Kanbar, Mohamed Ellabib, Hisham Al Jogol, Mohammed Muneer, Nuri Abdurraheim, Atirek Pratap Goel, Husham Abdelrahman, Hassan Al-Thani, Ayman El-Menyar

**Affiliations:** 1Department of Surgery, Orthopedic Surgery, Hamad General Hospital, Doha, Qatar; ims2311@rediffmail.com; 2Clinical Research, Trauma and Vascular Surgery, Hamad General Hospital, Doha, Qatar; masim1@hamad.qa; 3Department of Surgery, Trauma Surgery, Hamad General Hospital, Doha, Qatar; fmustafa2@hamad.qa (F.M.); akanbar@hamad.qa (A.K.); mellabib@hamad.qa (M.E.); hjogol@hotmail.com (H.A.J.); nabdurraheim@hamad.qa (N.A.); agoel@hamad.qa (A.P.G.); hushamco@hotmail.com (H.A.); 4Department of Surgery, Plastic Surgery, Hamad General Hospital, Doha, Qatar; mmyousif78@yahoo.com; 5Department of Surgery, Trauma and Vascular Surgery, Hamad General Hospital, Doha, Qatar; halthani@hamad.qa; 6Department of Clinical Medicine, Weill Cornell Medical College, Doha, Qatar

**Keywords:** femur fracture, orthopedic, trauma, management and outcome, Qatar

## Abstract

Background: Femur is the most fractured long bone in the body that often necessitates surgical fixation; however, data on the impact of the mechanism of injury (MOI), age, and timing of intervention are lacking in our region of the Arab Middle East. We aimed to describe the patterns, management, and outcome of traumatic femoral shaft fractures. Methods: A retrospective descriptive observational study was conducted for all trauma patients admitted with femoral shaft fractures between January 2012 and December 2015 at the only level 1 trauma center and tertiary hospital in the country. Data were analyzed and compared according to the time to intervention (intramedullary nailing; IMN), MOI, and age groups. Main outcomes included in-hospital complications and mortality. Results: A total of 605 hospitalized cases with femur fractures were reviewed. The mean age was 30.7 ± 16.2 years. The majority of fractures were unilateral (96.7%) and 91% were closed fractures. Three-fourths of fractures were treated by reamed intramedullary nailing (rIMN), antegrade in 80%. The pyriform fossa nails were used in 71.6% while trochanteric entry nails were used in 28.4%. Forty-five (8.9%) fractures were treated with an external fixator, 37 (6.1%) had conservative management. Traffic-related injuries occurred more in patients aged 14–30 years, whereas fall-related injuries were significantly higher in patients aged 31–59. Thirty-one patients (7.8%) had rIMN in less than 6 h post-injury, 106 (25.5%) had rIMN after 6–12 h and 267 (66.8%) had rIMN after more than 12 h. The implant type, duration of surgery, DVT prophylaxis, in-hospital complications, and mortality were comparable among the three treatment groups. Conclusions: In our center, the frequency of femoral fracture was 11%, and it mainly affected severely injured young males due to traffic-related collisions or falls. Further multicenter studies are needed to set a consensus for an appropriate management of femur fracture based on the MOI, location, and timing of injury.

## 1. Introduction

Traumatic femur fracture is a significant cause of morbidity with an annual incidence between 1.0 and 2.9 million worldwide [1]. Femur is the most fractured long bone in the body that often necessitates surgical fixation [2]. Femoral shaft fractures have bimodal distribution across different age groups with high velocity injuries, which are more common among adult males, while low energy injuries tend to be more common in children and elderly females [3]. The most frequently injured site of femur is the midshaft, particularly among adult population following road traffic collisions [4]. The pattern, presentation, and management of femoral fractures are influenced by the demographic characteristics, severity and mechanism of injury, and site of fracture [3,4,5]. The pattern of fracture varies owing to the direction of the force applied and the quantity of force absorbed during the trauma, and the aim of an early intervention is to get stable, anatomic fixation and to allow early mobilization [6]. 

Femur fractures might potentially result in short- or long-term disabilities, which can be measured, and their management thus reflects the effectiveness of the delivered trauma care during three intervals from the time of injury to the discharge. The intervals “between the time of trauma and hospital admission”, “from admission to surgical intervention”, and “from surgery to hospital discharge” were found to be easily measured and highly correlated to the known, accessible, and quantifiable data on health and economics [7]. However, these intervals vary from country to country and even within the same country according to the trauma system maturity, income, or resources and whether the injury was isolated or a polytrauma. 

Injuries associated with femoral shaft fractures are the major cause of morbidity in polytrauma patients [8,9]. There is evidence that femoral shaft fractures have frequent association with skeletal injuries (46.4%) and one-fourth of cases had occult associated injuries [10]. Notably, many studies have reported the presence of associated injuries such as hemothorax, bowel and head injuries concomitant with the femur shaft fracture, which reflect the severity of injury [10,11,12]. Of note, there are certain factors, such as age, gender, mechanism of injury, and magnitude of traumatic impact, which might influence the site of the femoral fracture [13,14]. Moreover, femoral fractures may be associated with severe complications, such as bleeding, pulmonary complications, deep vein thrombosis (DVT), and wound infection, in adult population [15].

There are various treatment options for femur fracture, such as conservative management, fixation with screw and plate, intramedullary nailing (IMN), open reduction and internal fixation (ORIF), and external fixation [3]. IMN is the gold standard treatment for femoral shaft fractures in adult patients [8]. Diaphyseal femur fractures are preferably treated with IMN, which helps to attain appropriate bone alignment; quicker bone healing that allows early mobilization, and lower rate of complications [16]. An earlier study from our center observed an association between the early IMN (within 12 h of injury) with fewer hospital complications and shorter length of hospital stay [5]. Similarly, Harvin et al. [15] reported an independent association between early IMN (within 24 h) for femur fixation and a lower rate of pulmonary complications. In Qatar, most victims of blunt trauma are young males sustaining proportionately higher injuries to the head, upper and lower extremities; only few articles address the femur injuries [17]. Herein, the present study aimed to assess the patterns, MOI, timing of management, and outcome of traumatic femoral fractures in the only level 1 trauma center and tertiary hospital in Qatar.

## 2. Materials and Methods

This retrospective study was conducted to include all trauma patients with femoral shaft fractures admitted at a level 1 trauma center at Hamad General Hospital (HGH) in Qatar between January 2012 and December 2015. During the study period, 6817 trauma cases were admitted at our referral center, which provides trauma services to the entire population (1,832,903 inhabitants in 2012 and 2,235,355 in 2015) in the state of Qatar. HGH is a modern, 600-plus bed facility that is considered one of the leading tertiary hospitals in the region. Hamad trauma center is the only tertiary facility that admits and treats patients with moderate to severe injury (on average, 1800 patients per year) free of charge for all the residents in the country.

Ethical approval was obtained from the Institutional Review Board (IRB#16240/16) of the Hamad Medical Corporation before commencing this study and the IRB has granted an exempt status for this retrospective study. This study followed the STROBE checklist (Supplementary table). All trauma patients admitted for the management of femoral fractures were included in the study. Patients with incomplete surgical data, treated without IMN, those who were brought in dead or died in the hospital before surgical treatment, and patients who were transferred to other facilities or travelled abroad for treatment were excluded from the study. Proximal and distal femur fracture patterns treated with other modalities were also excluded. 

Data were retrieved from the Qatar national trauma registry at HGH; therefore, they are national representative data as this is only level 1 trauma center in the country. Qatar Trauma Registry is a mature database that participates in both the National Trauma Data Bank and Trauma Quality Improvement Program of Committee on Trauma by the American College of Surgeons (TIQP-ACS). This trauma registry is well validated internally and externally on a regular basis. 

On arrival to the trauma center, all patients were evaluated and managed according to the advanced trauma life support (ATLS) guidelines. After clinical and radiological confirmation of the femur fracture, the orthopedic team intervened. All nails in the IMN cases were reamed (rIMN) and inserted in an antegrade or retrograde fashion in the lateral decubitus or supine position. Indications for retrograde nailing were ipsilateral acetabular, pelvis, or femoral neck fractures, polytrauma necessitating multiple simultaneous surgeries. Patients with initial external fixation underwent secondary IMN. 

Collected data included patient demographic characteristics, mechanism of injury, comorbidities, initial vital signs, associated injuries, abbreviated injury scores (AIS), injury severity score (ISS), Glasgow coma score (GCS), blood transfusion, DVT prophylaxis, pre-operative heparin, pattern of injuries (unilateral or bilateral), type of femur fracture (open/closed), management (conservative or IMN), time to intramedullary nailing, reamed, procedure (open/closed), site of entry (piriformis/trochanteric), implant type (antegrade/retrograde nail), in-hospital course, hospital length of stay, complications, and outcome. The AIS describes the severity of injuries at different body regions; the score ranges from 1 to 6. AIS scores of three most severely injured body regions were squared and added together to estimate the ISS, which provided an overall score for polytrauma (ISS 1–8 is minor injury, 9–15 is moderate, 16–24 is serious, 50–74 is critical, and 75 is non-survivable) [18]. The trauma registry prospectively records the fall data (fall from height and fall of heavy objects) using codes by International classification of diseases, 10th revision (ICD-10), which classified unintentional falls into 20 subcategories (W00-W19) [19]. 

Statistical Analysis: Data were reported as proportion, mean (± standard deviation), median, and range or interquartile range, when applicable. Group classification was performed according to our prior works [5,20]. Patients were categorized into three groups based on the time to intramedullary nailing (Group-I: < 6 h; Group-II: 6–12 h; and Group-III: >12 h). We have also analyzed the data according to the mechanism of injury and age groups (≤13 years, 14–30 years, 31–59 years, and ≥60 years). For each subgroup of patients, the chi-squared test was used to compare proportions between the categorical groups. Normality of continuous variables was checked by the Kolmogorov–Smirnov test. Continuous variables were compared using Student’s *t*-test for two groups or one-way ANOVA test for over two groups, for parametric data. Yates’ corrected chi-square was used for categorical variables if the expected cell frequencies were below five. A two-tailed *p* value of <0.05 was considered statistically significant. Data analysis was carried out using SPSS, version 18 (SPSS Inc., Chicago, IL, USA).

## 3. Results

During the 4-year period, 6817 trauma patients were admitted to our center, and 740 (10.9%) of them presented with a femur fracture. As many as 104 cases were excluded because of incomplete information, 18 patients died in the hospital prior to intervention, 10 were transferred to another facility, and 3 were brought in dead to the hospital. Thus, after excluding 135 patients, 605 patients (9%) were analyzed and they constituted the study cohort (Figure 1). 

The majority of the study cohort were males (89.4%) with the mean age of the cohort of 30.7 ± 16.2 years. Table 1 shows that most fractures (*n* = 393, 65%) resulted from road traffic collisions, followed by fall from height (26%), and fall of a heavy object (6%). The frequent comorbidities were hypertension (7%), diabetes mellitus (6.4%), and bronchial asthma (6.0%). The mean initial vital signs such as heart rate, body temperature, systolic blood pressure, and respiratory rate were unremarkable in the study cohort. The mean ISS was 14.8 ± 8.1. Associated injuries to the chest, head, and abdomen were found in 24.1%, 16.2%, and 15.9% cases, respectively; 43% had concomitant lower extremity fracture requiring internal fixation; 18% had associated pelvis fracture; and 19.8% sustained spine fracture. 

Table 2 shows the management, complications, and outcome of the femur fractures. Deep vein thrombosis (DVT) prophylaxis was given to 76.5% of the patients and only 7.3% received pre-operative heparin. Most femur fractures were unilateral (96.7%) and 91% were closed fractures. Three-fourths of the femur fractures were treated by reamed intramedullary nailing (rIMN), antegrade in 80% and retrograde in 20% cases. The pyriform fossa nails (71.6%) were frequently used, followed by trochanteric entry nails (28.4%). Forty-five (8.9%) of the fractures were treated with an external fixator, and 37 (6.1%) had conservative management. Blood transfusion was required in 39.5% cases, with a median of four blood units. Post-treatment, 12.6 % of patients developed wound infection (infections were treated with local wound care, debridement, nail removal, and delayed exchange nailing, whenever needed). About 9.7% of them were diagnosed with pulmonary complications, such as pneumonia (7.3%), pulmonary embolism (1.2%), and acute respiratory distress syndrome (1.2%). The other complications included sepsis (4.0%) and acute renal failure secondary to acute tubular necrosis (2.1%). The median lengths of hospital and ICU stays were 10 days and 7 days, respectively. The overall hospital mortality rate was 2.1%.

Table 3 shows an analysis of the management, complications, and outcome based on the timing of intramedullary nailing. Patients were divided into three groups: 31 (7.8%) were in Group I (rIMN < 6 h), 106 (25.5%) were in Group II (rIMN 6–12 h), and 267 (66.8%) were in Group III (rIMN > 12 h). In Groups I and II, the common site of entry was piriformis, whereas trochanteric entry was more evident in Group III patients (*p* = 0.009). The implant type, duration of surgery, DVT prophylaxis, in-hospital complications, and mortality were comparable among these three groups.

Table 4 shows the clinical characteristics, management, and outcome of femur fracture based on the mechanisms of injury. In comparison with the other groups, victims of traffic-related collisions were younger, sustained severe injuries (mean ISS; 15.8 ± 8.7; *p* = 0.001), and frequently had associated tibia (*p* = 0.006) and fibula fractures (*p* = 0.02). Associated abdominal injuries were observed more often in patients injured by a fall of a heavy object (*p* = 0.03). The rate of rIMN was significantly higher in victims of traffic collisions (*p* = 0.005). Patients who sustained a fall of a heavy object were more likely to receive conservative treatment. The need for blood transfusion was significantly higher in victims of traffic collisions and of a fall of a heavy object (*p* = 0.002). There was no significant difference with respect to the time of rIMN, site of entry, implant type, procedure, in-hospital complication, and mortality among all the groups.

Table 5 shows the predominance of the age groups 14–30 years (46.4%) and 31–59 years (36.9%). Moreover, male patients predominated in the age group 31–59 years, compared with other groups. Traffic-related injuries were more frequent in patients aged 14–30, whereas injuries related to a fall from height and fall of a heavy object were significantly more frequent in patients aged 31–59 (Figure 2). Young adults, aged 14–30, sustained severe injuries with higher injury severity scores (16.2 ± 8.9; *p* = 0.001), had frequent spinal injuries (*p* = 0.004), and underwent rIMN (*p* = 0.001) as compared with the other groups. Patients aged ≤13 years were frequently managed conservatively. Patients aged 31–59 were more likely to receive DVT prophylaxis (*p* = 0.001) and had higher rate of wound infection (*p* = 0.02). The need for blood transfusion (*p* = 0.001) and prolonged hospitalization (*p* = 0.001) were evident among elderly population (≥60 years) when compared with other age groups. The study groups did not differ significantly with respect to the other associated injuries, in-hospital complications, and mortality.

## 4. Discussion

It is imperative to assess the clinico-epidemiological characteristics, mechanisms of injury, and patterns of femoral fractures to review the appropriateness of management practices and develop preventive measures. There are several key features of the current analysis. It was identified that the proportion of traumatic femur fracture cases managed at our center was 11%, which is in agreement with an earlier study that reported similar rates of femoral fractures treated at a regional trauma center in South Nigeria [21]. The results of our study showed a preponderance of male gender (89.4%) and young age (mean: 30 years), which is in line with an earlier study from Saudi Arabia, but the mean age was a slightly lower than the mean of 33 years reported there [22]. In contrast, Khan et al. [3] reported higher proportion of females (58%) and advanced age (mean: 63 years) among patients with distal femoral shaft fractures treated at a tertiary referral hospital in London. An earlier study from Saudi Arabia reported that half of femur fracture victims belonged to the age group of 16–30 followed by 30–60 (39.3%) [22]. The present study also showed a predominance of the ages of 14–30 (46.4%) and 31–59 (36.9%), which agrees with the Saudi Arabia study and could be due to the sociodemographic similarity between the two neighboring countries. As 39.5% and 52.8% of femur fracture patients aged 31–59 sustained a fall from height and a fall of a heavy object, respectively, these could have been work-related injuries, such as during construction work, or domestic falls. This finding needs further studies for better elaboration and causation.

Road traffic collisions (65%) remain the most common injury mechanism followed by fall from height in our series. These findings agree with earlier studies which also reported high impact trauma, mainly road traffic crashes as the commonest cause of femur fractures [2,21]. Furthermore, traffic-related injuries were more evident among young individuals of 14–30 years of age, who represent the most active age group of young individuals, usually involved in over-speeding and reckless driving. In our study group, victims of traffic-related collisions had associated fractures of tibia and fibula and more likely underwent rIMN. Similarly, predominance of femoral shaft fractures secondary to road traffic collisions among younger males was reported by an earlier study from Romania [23]. A recent study on the management of femoral shaft fractures reported road traffic collisions as the commonest cause and suggested interlocking intramedullary nailing as the modality of choice for candidates requiring operative intervention [24]. In our study, the thorax, head, and abdomen were the frequently associated injured body regions. This finding was supported by a recent meta-analysis that reported high-energy trauma as the major cause of femur fracture with concomitant injuries to the chest and head regions [20]. 

In our series, most femur fractures were closed and unilateral. The pattern of closed femoral fractures is frequently observed due to soft tissue cover of the femur, which in contrast with the tibial fractures [10]. Ibeanusi and Chioma [21] reported higher proportion of femoral fractures to be closed (78%), compared with open fractures (22%), which were more likely to involve diaphyseal femur fracture (58.1%) secondary to high impact trauma by road traffic collisions or gunshot injuries. On the other hand, open femur fractures are not uncommon and range from 16.5 to 23% [25,26]. An external fixator construct could be used to stabilize hemodynamically unstable patients or those with severe open fractures, in accordance with the recommendations for safely performing IMN in lower grade open femoral shaft fractures [27,28]. Reamed intramedullary nailing is the standard of care treatment in our institution for shaft fractures of long bones, particularly the closed method, which has been suggested as superior to other procedures, despite the controversy [29,30,31]. 

In our series, the median time to stabilize femur fractures by rIMN was within the first 20 hours of admission, but early stabilization was observed in 34.3% of cases as opposed to 51% reported in earlier studies from our center [5,20]. The earlier studies advocated the beneficial effect of early definitive fixation of femur fractures within 24 hours among patients suitable for IMN [15,32]. Of note, the type of implant, duration of surgery, DVT prophylaxis, and in-hospital complication did not differ significantly based on the time to rIMN in the present study. This could be due to the fact that we have categorized the time of rIMN as very early, early, and delayed surgery but still demonstrated that two-thirds of the patients were treated after 12 hours of admission.

Antegrade nailing was performed in most of our cases with lateral positioning without the use of a fracture table, as described by Bishop et al. [33]. An earlier study by Wolinsky et al. [34] suggested a significant decrease in the operating times with this technique; however, the current study did not compare operating times based on positioning (lateral versus supine). In about one-fourth of fractures, a retrograde nail was inserted in the supine position, which is in accordance with the indications described by sanders et al. [35]. Surgeons’ preference in our series was antegrade nails involving entry from pyriform fossa in the majority (72%) of cases, and the remaining cases had trochanteric entry nails. In contrast to our practice, a systematic review by Kumar et al. [36] identified trochanteric entry nailing to be superior to pyriform fossa nailing to treat femur shaft fractures in adults. The authors also suggested the ease of learning the technique of entry through greater trochanteric tip that resulted in improved functional outcomes, although there is no difference in the rate of union among the two entry sites. Another prospective cohort study on antegrade femoral nailing reported similar higher rate of union, lower complication rate, and comparable functional results of trochanteric insertion as compared with the piriformis fossa nailing [37]. In addition, the authors demonstrated lesser fluoroscopy and operation time with greater trochanter entry in obese patients. Further supporting the notion of trochanteric entry nails, another study on cadavers found lesser structural and iatrogenic injury to the surrounding structures and the gluteal musculature with trochanteric nailing [38]. In our series, open reduction was performed in about one-third of patients and this approach was secondary to difficult closed reduction procedure, which has consistently been described as a safe alternative technique [39,40]. In this study, external fixation for the treatment of femoral shaft was primarily performed for open fractures. Although this is not a standard technique to treat femoral shaft fracture, exceptionally, it can be used to manage open fractures with concomitant complex soft tissue injuries [21].

In this study, about 40% of patients required blood transfusion, which is markedly less than the reported incidence in an earlier study [41]. This is attributed to the fact that after the initial resuscitation in polytrauma patients, we meticulously avoid blood transfusions when hemoglobin levels are higher than 8 mg/dL [42] in asymptomatic patients who responded well to physiotherapy and ambulation efforts. 

The rate of surgical site infection after IMN was reported to be 11.8% for combined femur and tibial fractures [43], whereas the overall rate of infection for isolated femur fractures was found to be as low as 0.8% [44]. Notably, the rate of wound infection in our series was higher and all patients with surgical site infection were managed with local wound care and antibiotics, implant retention and did not require surgical debridement or implant removal. An earlier study, which analyzed the outcome of femoral fractures, reported a lower infection rate (5.4%) as compared with our findings [21].

In our cohort, the rate of pulmonary embolism and acute respiratory distress syndrome (ARDS) was found to be 1.2%. Similar to our observations, Kim et al. [45] reported a slightly higher frequency of pulmonary embolism (2.2%), which was developed soon post-trauma. Bosse et al. [46] reviewed femur fractures management at two different settings: one center used rIMN and the other one mainly used plates to treat femoral fractures. The authors suggested that rIMN of femoral fractures did not increase the risk of pulmonary complications and there was no significant difference between the two cohorts with respect to the incidence of pulmonary complication and mortality. Other studies have also reported that hemodynamically stable patients with pulmonary injuries and femoral fracture can be successfully treated with rIMN [47,48]. Moreover, a recent meta-analysis identified that early IMN has lower risk of pulmonary complications such as ARDS and pneumonia compared with delayed IMN fixation [20]. Therefore, in polytrauma patients, the reported pulmonary complications might be associated with thorax trauma rather than the IMN [49].

There are certain limitations to this study owing to the retrospective study design and data retrieval from registry database, that is, some variables with incomplete information. We excluded 138 cases with femur fractures, as shown in Figure 1, and this could have underestimated the frequency and outcome of the use of rIMN as well as the selection bias. This study did not address the functional outcome as radiological union (nonunion, malunion, extended delayed union) and clinical follow-up details about physical therapy, early mobilization, and counseling. As the passage of time is a potential source of bias, it may be relevant to note that the main changes in the last five years in our institution included the damage control concept in polytrauma; more external fixator use, massive transfusion protocol, and all the used IMN are reamed (rIMN). However, these data were abstracted from the Qatar national trauma registry and were validated internally and externally on regular basis. The study results could be representative of the surgically treated femur fractures in the country.

## 5. Conclusions

The frequency of femoral fracture is 11% in our center, which are mainly encountered in severely injured young males and caused by traffic-related collisions and falls from height. The femoral fracture represents a spectrum of injury characteristics, from simple isolated injuries, requiring a simple intramedullary nail, to polytrauma patients with associated injuries that require a multidisciplinary treatment approach. We believe that a clinico-epidemiological study may help surgeons to understand the pattern of fractures, management, and complications to improve patients’ outcomes. Our findings may help healthcare policy prioritization, resource allocation planning, and implementation of the best practice. Further multicenter studies are needed to reach a consensus for the appropriate management based on the location and timing of injury.

## Figures and Tables

**Figure 1 ijerph-18-05916-f001:**
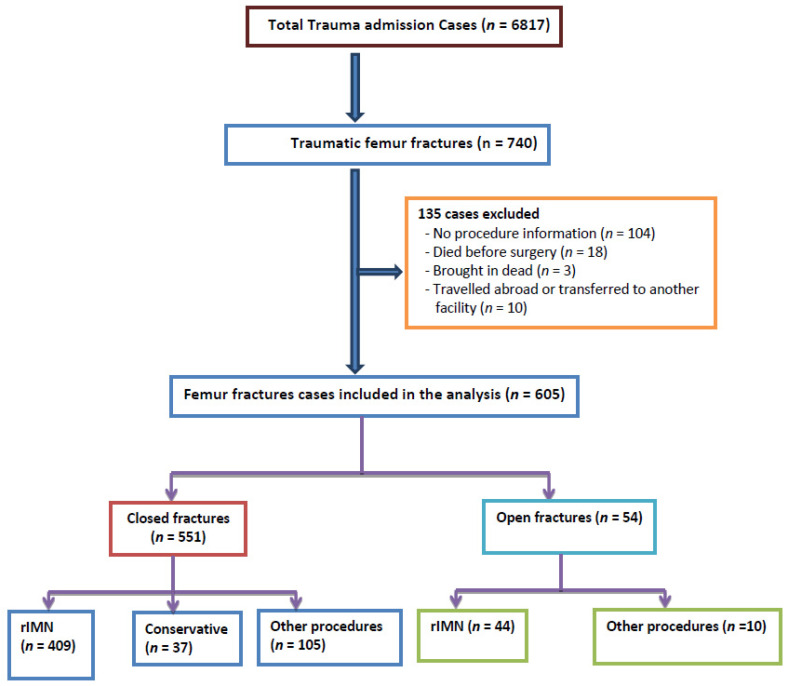
Study design. rIMN, reamed intramedullary nailing.

**Figure 2 ijerph-18-05916-f002:**
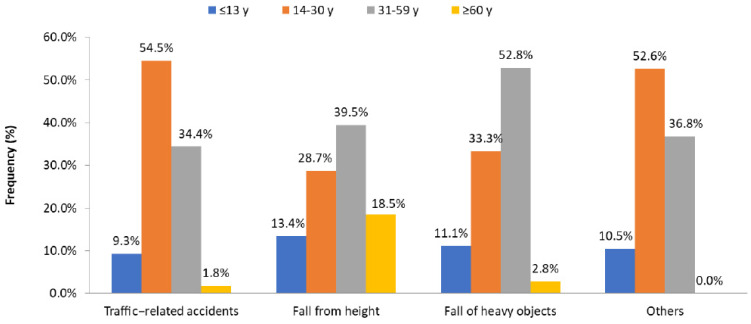
Mechanism of injury in patients with femur fractures by age group.

**Table 1 ijerph-18-05916-t001:** Demographics and clinical characteristics of patients with traumatic femur fracture (*n* = 605).

Variable	Value	Variable	Value
**Age (mean ± SD)**	30.7 ± 16.2	**Associated injuries**	
**Males**	541 (89.4%)	Tibia	106 (17.5%)
**Mechanism of Injury**		Fibula	68 (11.2%)
MVC	264 (43.6%)	Ankle	46 (7.6%)
Pedestrian	81 (13.4%)	Knee	39 (6.4%)
Motorcycle/bike crash	32 (5.3%)	Pelvis	109 (18.0%)
ATV	16 (2.6%)	Head	98 (16.2%)
Fall from height	157 (26.0%)	Chest	146 (24.1%)
Fall of a heavy object	36 (6.0%)	Abdomen	96 (15.9%)
Others	19 (3.1%)	Spine	120 (19.8%)
**Comorbidities**		**Initial lab results**	
Hypertension	42 (6.9%)	Hemoglobin level (*n* = 509)	13.0 ± 2.2
Diabetes mellitus	39 (6.4%)	White blood cell count (*n* = 488)	15.2 ± 6.4
Bronchial asthma	36 (6.0%)	Neutrophil count (*n* = 255)	14.7 ± 13.8
Coronary artery disease	14 (2.3%)	Platelet count (*n* = 499)	259.5 ± 92.3
**Initial assessment**		International normalized ratio (*n* = 466)	1.07 ± 0.14
Initial heart rate	96.3 ± 21.4		
Initial SBP	125.9 ± 20.4		
Respiratory rate	20.3 ± 5.9		
Injury severity score	14.8 ± 8.1		

MVC: motor vehicle crash, ATV: all-terrain vehicle, SBP: systolic blood pressure.

**Table 2 ijerph-18-05916-t002:** Management, complications, and outcome of femur fractures.

Variable	Value
**DVT prophylaxis (*n* = 489)**	374 (76.5%)
**Pre-operative heparin (*n* = 410)**	30 (7.3%)
**Unilateral femur fracture**	584 (96.7%)
**Bilateral femur fracture**	20 (3.3%)
**Type of Femur Fracture**	
Close fracture	551 (91.1%)
Open fracture	54 (8.9%)
**Reamed intramedullary nailing**	453 (74.9%)
**Time to IMN (hours) (*n* = 400)**	20 (1–4382)
Early (≤12 h)	137 (34.3%)
Late (>12 h)	267 (66.8%)
**External fixation**	45 (8.9%)
**Conservative management**	37 (6.1%)
**Implant type (*n* = 408)**	
Antegrade nail	327 (80.1%)
Retrograde nail	81 (19.9%)
**Site of entry (*n* = 342)**	
Piriformis	245 (71.6%)
Trochanteric	97 (28.4%)
**Procedure (*n* = 439)**	
Open	145 (33.0%)
Closed	294 (67.0%)
**Duration of Surgery (hours)**	2.6 ± 1.7
**Locking**	453 (100%)
**Number of procedures (*n* = 423)**	1 (1–7)
**Bleeding on admission (*n* = 432)**	38 (8.8%)
**Wound Infection (*n* = 461)**	58 (12.6%)
**Blood transfusion**	239 (39.5%)
**Blood units transfused**	4 (1–51)
**Complications**	
Pneumonia	44 (7.3%)
Pulmonary embolism	7 (1.2%)
Acute respiratory distress syndrome	7 (1.2%)
Sepsis	24 (4.0%)
Acute renal failure	13 (2.1%)
Hospital length of stay in days	10 (1–157)
ICU stay in days	7 (1–88)
**Mortality**	13 (2.1%)

**Table 3 ijerph-18-05916-t003:** Management, complications, and outcome by timing of intramedullary nailing (IMN).

Variable	Time to Intramedullary Nailing *			*p* Value
	Group I: < 6 h (*n* = 31; 7.7%)	Group II: 6–12 h (*n* = 106; 26.2%)	Group III: >12 h (*n* = 267; 66.1%)	
**Timing to IMN, median (IQR) h**	4.5 (4.0–5.5)	10 (8.4–11.0)	24 (20–72)	0.001
**Site of entry (*n* = 301)**				
Piriformis	23 (82.1%)	70 (82.4%)	124 (66.0%)	0.009 for all
Trochanteric	5 (17.9%)	15 (17.6%)	64 (34.0%)	
**Implant type (*n* = 364)**				
Antegrade nail	26 (96.3%)	76 (77.6%)	186 (77.8%)	0.07 for all
Retrograde nail	1 (3.7%)	22 (22.4%)	53 (22.2%)	
**Duration of Surgery (h)**	3.2 ± 1.6	2.7 ± 1.6	2.6 ± 1.5	0.17
**Complications**				
Wound Infection (*n* = 312)	3 (12.5%)	12 (13.3%)	27 (13.6%)	0.97
DVT prophylaxis (*n* = 320)	19 (79.2%)	67 (79.8%)	159 (75.0%)	0.65
Sepsis	2 (6.5%)	3 (2.9%)	9 (3.4%)	0.91
Pulmonary Embolism	0 (0.0%)	0 (0.0%)	6 (2.2%)	0.52
Acute Respiratory Distress Syndrome	1 (3.2%)	1 (1.0%)	3 (1.1%)	0.96
Pneumonia	2 (6.5%)	4 (3.9%)	21 (7.9%)	0.51
**Mortality**	0 (0.0%)	1 (1.0%)	4 (1.5%)	0.96

* Data were available for 404 cases, IQR: interquartile range, h: hour.

**Table 4 ijerph-18-05916-t004:** Management, complications, and outcome by mechanism of injury.

Variable	Traffic-Related Injury (*n* = 393)	Fall from Height (*n* = 157)	Fall of Heavy Object (*n* = 36)	*p* Value
**Age (mean ± SD)**	27.8 ± 12.7	37.9 ± 21.6	31.7 ± 13.9	0.001
**Males**	359 (91.3%)	132 (84.1%)	34 (94.4%)	0.05
**Injury Severity Score (ISS)**	15.8 ± 8.7	12.9 ± 6.3	14.1 ± 6.5	0.001
**ISS > 15**	138 (35.1%)	39 (24.8%)	11 (30.6%)	0.02
**Associated injuries**				
Tibia	83 (21.1%)	14 (8.9%)	7 (19.4%)	0.006
Fibula	55 (14.0%)	11 (7.0%)	1 (2.8%)	0.02
Ankle	31 (7.9%)	9 (5.7%)	3 (8.3%)	0.44
Knee	29 (7.4%)	8 (5.1%)	1 (2.8%)	0.59
Pelvis	72 (18.3%)	23 (14.6%)	10 (27.8%)	0.29
Head	70 (17.8%)	23 (14.6%)	3 (8.3%)	0.37
Chest	99 (25.2%)	33 (21.0%)	10 (27.8%)	0.69
Abdomen	70 (17.8%)	14 (8.9%)	7 (19.4%)	0.03
Spine	80 (20.4%)	26 (16.6%)	10 (27.8%)	0.46
**Management**				
Reamed IMN	307 (78.1%)	102 (65.0%)	28 (77.8%)	0.005 for all
Conservative	16 (4.1%)	15 (9.6%)	5 (13.9%)	
Other procedures	70 (17.8%)	40 (25.5%)	3 (8.3%)	
**Time to IMN (hours)**				
Early (<12 h)	76 (28.6%)	23 (24.7%)	8 (30.8%)	0.88
Late (≥12 h)	190 (71.4%)	70 (75.3%)	18 (69.2%)	
**Site of entry**				
Piriformis	166 (72.2%)	53 (67.1%)	16 (76.2%)	0.60
Trochanteric	64 (27.8%)	26 (32.9%)	5 (23.8%)	
**Implant type**				
Antegrade nail	215 (77.6%)	79 (84.9%)	21 (84.0%)	0.27
Retrograde nail	62 (22.4%)	14 (15.1%)	4 (16.0%)	
**Procedure**				
Open	196 (66.4%)	70 (69.3%)	18 (66.7%)	0.93
Closed	99 (33.6%)	31 (30.7%)	9 (33.3%)	
**Complications**				
Wound Infection	46 (15.1%)	8 (6.7%)	2 (8.3%)	0.11
DVT prophylaxis	249 (78.5%)	95 (72.5%)	21 (72.4%)	0.53
Blood transfusion	172 (43.8%)	49 (31.2%)	16 (44.4%)	0.002
Blood units transfused	4 (1–51)	2 (1–22)	2 (1–16)	0.009
Sepsis	19 (4.8%)	3 (1.9%)	2 (5.6%)	0.31
Pulmonary embolism	3 (0.8%)	4 (2.5%)	0 (0.0%)	0.28
Acute respirtory distresss syndrome	6 (1.5%)	1 (0.6%)	0 (0.0%)	0.68
Pneumonia	35 (8.9%)	4 (2.5%)	3 (8.3%)	0.06
**Hospital length of stay**	10 (1–157)	9 (1–117)	11 (3–104)	0.07
**Mortality**	12 (3.1%)	1 (0.6%)	0 (0.0%)	0.21

There were 19 cases with other mechanisms of injury.

**Table 5 ijerph-18-05916-t005:** Clinical characteristics, management, complications, and outcome by age groups (*n* = 601).

Variable	≤13 Years (*n* = 63; 10.5%)	14–30 Years (*n* = 279; 46.4%)	31–59 Years (*n* = 222; 36.9%)	≥60 Years (*n* = 37; 6.2%)	*p* Value
Age (mean ± SD) years	5.9 ± 3.7	23.3 ± 4.5	40.1 ± 7.4	71.9 ± 7.5	0.001
**Males**	50 (79.4%)	256 (91.8%)	211 (95.0%)	20 (54.1%)	0.001
**ISS** mean ± SD	12.2 ± 6.1	16.2 ± 8.9	14.2 ± 7.2	12.8 ± 7.7	0.001
**ISS** >15	10 (15.9%)	104 (37.3%)	66 (29.7%)	9 (24.3%)	0.005
**Associated injuries**					
Tibia	5 (7.9%)	48 (17.2%)	47 (21.2%)	5 (13.5%)	0.09
Fibula	4 (6.3%)	28 (10.0%)	32 (14.4%)	3 (8.1%)	0.20
Ankle	3 (4.8%)	22 (7.9%)	19 (8.6%)	1 (2.7%)	0.50
Knee	5 (7.9%)	17 (6.1%)	15 (6.8%)	2 (5.4%)	0.94
Pelvis	10 (15.9%)	55 (19.7%)	38 (17.1%)	3 (8.1%)	0.34
Head	6 (9.5%)	49 (17.6%)	38 (17.1%)	3 (8.1%)	0.22
Chest	11 (17.5%)	73 (26.2%)	57 (25.7%)	4 (10.8%)	0.11
Abdomen	10 (15.9%)	53 (19.30%)	31 (14.0%)	1 (2.7%)	0.05
Spine	4 (6.3%)	66 (23.7%)	46 (20.7%)	3 (8.1%)	0.004
**Management**					
Reamed IMN	37 (58.7%)	234 (83.9%)	161 (72.5%)	18 (48.6%)	0.001 for all
Conservative	17 (27.0%)	7 (2.5%)	10 (4.5%)	3 (8.1%)	
Other procedures	9 (14.3%)	38 (13.6%)	51 (23.0%)	16 (43.2%)	
**Time to IMN (hours)**					
Early (<12 h)	8 (24.2%)	55 (26.7%)	44 (30.6%)	4 (25.0%)	0.81 for all
Late (≥12 h)	25 (75.8%)	151 (73.3%)	100 (69.4%)	12 (75.0%)	
**Site of entry**					
Piriformis	16 (57.1%)	135 (75.4%)	87 (71.3%)	5 (50.0%)	0.09 for all
Trochanteric	12 (42.9%)	44 (24.6%)	35 (28.7%)	5 (50.0%)	
**Implant type**					
Antegrade nail	22 (64.7%)	176 (83.0%)	113 (77.9%)	14 (93.3%)	0.04 for all
Retrograde nail	12 (35.3%)	36 (17.0%)	32 (22.1%)	1 (6.7%)	
**Complications**					
Wound Infection	2 (4.3%)	26 (12.6%)	30 (16.8%)	0 (0.0%)	0.02
DVT prophylaxis	14 (28.0%)	181 (80.8%)	154 (84.2%)	22 (75.9%)	0.001
Blood transfusion	7 (11.1%)	122 (43.7%)	86 (38.7%)	22 (59.5%)	0.001
Blood units transfused	2 (1–10)	4 (1–51)	4 (1–24)	3 (1–8)	0.03
Sepsis	1 (1.6%)	12 (4.3%)	8 (3.6%)	3 (8.1%)	0.43
Pulmonary Embolism	2 (3.2%)	3 (1.1%)	1 (0.5%)	1 (2.7%)	0.26
Acute respiratory distress syndrome	0 (0.0%)	4 (1.4%)	3 (1.4%)	0 (0.0%)	0.70
Pneumonia	3 (4.8%)	21 (7.5%)	19 (8.6%)	1 (2.7%)	0.51
**Hospital length of stay**	6 (1–43)	10 (1–157)	10 (1–123)	15 (4–115)	0.001
**Mortality**	0 (0.0%)	7 (2.5%)	3 (1.4%)	1 (2.7%)	0.50

ISS = Injury Severity Score

## Data Availability

Not applicable.

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
