# Peer review of "Patterns, Management, and Outcome of Traumatic Femur Fracture: Exploring the Experience of the Only Level 1 Trauma Center in Qatar"

_ijerph, 2021, doi:10.3390/ijerph18115916_

Round 1

Reviewer 1 Report

This is a well written manuscript on femur fractures

The English language and clarity of the Introduction could be improved.

The selection criteria and in particular the excluded cases should be better expressed (eg died before surgery includes "brought in dead"; patients not treated in the center could substitute "travelled outside...")

Author Response

Comments and Suggestions for Authors

This is a well written manuscript on femur fractures

The English language and clarity of the Introduction could be improved.

The selection criteria and in particular the excluded cases should be better expressed (eg died before surgery includes "brought in dead"; patients not treated in the center could substitute "travelled outside...")

Reply: thanks. Selection criteria revised and English language and clarity of introduction improved

Reviewer 2 Report

The manuscript “Patterns, Management and Outcome of Traumatic Femur Fracture: Exploring the Experience of the only Level 1 Trauma Center in Qatar” by Syed Imran Ghouri, Mohammad Asim, Fuad Mustafa, Ahad Kanbar, Mohamed Ellabib, Hisham Al Jogol, Mohammed Muneer, Nuri Abdurraheim, Atirek Pratap Goel, Husham Abdelrahman, Hassan Al-Thaniand Ayman El-Menyar  is an article that aimed  to assess the patterns, associated injuries, timing of management and out- come of traumatic femoral fractures in the only level 1 trauma center and tertiary hospital in Qatar.

Below are my comments and remarks regarding the article:

Introduction

  1. Method of femur fixation is TKA?

Methods:

There is no description of the injury severity score ISS used

Quite a large proportion of patients excluded due to "No procedure information"

Retrospective poorly planned study. Specific goals should be set from the outset. Lots of data, but few concrete results.

Results

  1. Table 1. MVC ATV abbreviations are not explained
  2. Table 1 is not readable as unrelated variables are presented next to each other
  3. Group II (rIMN> 12 h) should be Group III
  4. Time to intramedullary nailing * why again the division into 3 groups instead of giving specific numbers (hours) and their comparison in groups
  5. Table 5 is similar to the above. Why such and not others division into age groups (≤13 years, 14-30 years, 31-59 years and ≥60 years) and why the average age was not simply compared.

Discussion

  1. Epidemiology, even too widely discussed, sometimes there is no polemics as to why the data differ from other authors
  2. Brumbacks and colleagues probably supposed to be Brumbacks et al.

Editors:

No consequences of using INM (intramedullary nailing) abbreviations

Author Response

Introduction

Method of femur fixation is TKA?

Reply: sentence revised

Methods:

There is no description of the injury severity score ISS used

Reply: description added

Quite a large proportion of patients excluded due to "No procedure information"

Retrospective poorly planned study. Specific goals should be set from the outset. Lots of data, but few concrete results.

Reply: we clearly mentioned the numbers and reasons for excluded cases in the text and in figure 1, this study has specific goals (at the end of the introduction) and also limitations as a retrospective study

Results

Table 1. MVC ATV abbreviations are not explained

Reply: thanks, we spelled it out in the table

Table 1 is not readable as unrelated variables are presented next to each other

Reply: we revised the headings and subheadings in table 1 to be clearer

Group II (rIMN> 12 h) should be Group III

Reply: thanks, corrected

Time to intramedullary nailing * why again the division into 3 groups instead of giving specific numbers (hours) and their comparison in groups

Reply: we corrected the IMN groups in table 3

Table 5 is similar to the above. Why such and not others division into age groups (≤13 years, 14-30 years, 31-59 years and ≥60 years) and why the average age was not simply compared.

Reply: I am so sorry; I could not get you, could you clarify your comment again. By the way, we used the first age group<=13 years old as pediatric group as per our hospital protocol

Discussion

Epidemiology, even too widely discussed, sometimes there is no polemics as to why the data differ from other authors

Reply: we addressed the similarity and difference between our study and others especially the involvement of younger patients in our cohort and pointed to the impact of timing, mechanism of injury and age

Brumbacks and colleagues probably supposed to be Brumbacks et al.

Reply: thanks, revised

Editors:

No consequences of using INM (intramedullary nailing) abbreviations

Reply: we did not use INM but we used IMN for intramedullary nailing and rIMN for reamed IMN (if authors clearly mentioned reamed we used rIMN, otherwise we used IMN)

Reviewer 3 Report

Thanks for the opportunity to review this manuscript. This is a retrospective cohort study that examined the patterns, management, and outcome (e.g., mortality) of traumatic femoral shaft fractures from a level-1 trauma center in Qatar. Results were well presented. Overall, the manuscript is well written. There are strengths and weaknesses in this study. My following comments include a summary of weaknesses and offer suggestions for the authors' consideration.

  1. In Abstract, the background section only included the aim. Please add a rationale of why this study is important.
  2. In Introduction Line 73-74, the authors stated that “The pattern, presentation and management of femoral fractures are influenced by demographic characteristics, severity and mechanism of injury, and site of fracture”. Please add the references. In addition, please elaborate a bit more on this statement to provide stronger rationales for your study purposes.
  3. This study was conducted at the Hamad General Hospital (HGH). What is the size of the hospital? This information may be useful for other similar size hospitals.
  4. There are some grammar errors in this manuscript: for example, Line 96 “it is regularly reporting to the national trauma databank at USA” should be “regularly reported to…in the USA; Line 178 “were more likely to be receive conservative treatment” should be “more likely to receive…”. Please carefully check the manuscript throughout and correct those errors.
  5. Line 107, how the “mechanism of injury” was collected? Was the data collection method reliable? Please clarify.
  6. For the group classification that was based on the time to treatment and age, is there any rationale? Please clarify.
  7. I assume that the “ANOVA test” (Line 123) referred to one-way ANOVA. Were post-hoc comparisons also conducted?
  8. For participant selection, 135 cases (18%) were excluded (Figure 1). Would this lead to over-estimate or under-estimate of the results? Please comment on this in the limitations.
  9. The “fall from height” and “fall of heavy” were not clearly defined in the methods. Please provide the definitions and examples.
  10. Line 188, “injuries related to fall from height and fall of heavy object were significantly higher in patients aged 31-59 years”. Since the age range is relatively wide from the young to mid-age. I wonder if those injuries are related to occupations such as construction work.
  11. Elderly people (≥60 years) usually fall more often than the younger groups, and may suffer hip fracture (femur head or neck). I wonder if there are similar findings from this study. Meanwhile, I wonder why only 37 older adults were included in this study.
  12. The authors did a good job in the Discussion. However, in the limitations, several issues may need to be mentioned: (1) representative of the results as they are from one centre (hospital); (2) reliability of the data collection (e.g., injury mechanism); and (3) lack of post-treatment outcomes, which are important to guide the injury management.

Author Response

Thanks for the opportunity to review this manuscript. This is a retrospective cohort study that examined the patterns, management, and outcome (e.g., mortality) of traumatic femoral shaft fractures from a level-1 trauma center in Qatar. Results were well presented. Overall, the manuscript is well written. There are strengths and weaknesses in this study. My following comments include a summary of weaknesses and offer suggestions for the authors' consideration.

In Abstract, the background section only included the aim. Please add a rationale of why this study is important.

Reply: we added background

In Introduction Line 73-74, the authors stated that “The pattern, presentation and management of femoral fractures are influenced by demographic characteristics, severity and mechanism of injury, and site of fracture”. Please add the references. In addition, please elaborate a bit more on this statement to provide stronger rationales for your study purposes.

Reply: thanks, we added references and more elaboration in the introduction for better clarification

This study was conducted at the Hamad General Hospital (HGH). What is the size of the hospital? This information may be useful for other similar size hospitals.

Reply: we added some info in this regard

There are some grammar errors in this manuscript: for example, Line 96 “it is regularly reporting to the national trauma databank at USA” should be “regularly reported to…in the USA; Line 178 “were more likely to be receive conservative treatment” should be “more likely to receive…”. Please carefully check the manuscript throughout and correct those errors.

Reply: Thanks, errors have been corrected

Line 107, how the “mechanism of injury” was collected? Was the data collection method reliable? Please clarify.

Reply: the mechanism of injury and other data were collected from the Qatar national trauma registry which has internal and external validation on regular basis and it is linked to the National Trauma Data Bank (NTDB) which is the largest aggregation of U.S. trauma registry data ever assembled. Qatar Trauma Registry is a mature database that participates in both National Trauma Data Bank and Trauma Quality Improvement Program of Committee on Trauma by the American College of Surgeons (TIQP-ACS).

For the group classification that was based on the time to treatment and age, is there any rationale? Please clarify.

Reply: the group classification was based on our prior works: El-Menyar A, Muneer M, Samson D, et al. Early versus late intramedullary nailing for traumatic femur fracture management: meta-analysis. J Orthop Surg Res. 2018;13(1):160. doi: 10.1186/s13018-018-0856-4.

Alobaidi AS, Al-Hassani A, El-Menyar A, et al. Early and late intramedullary nailing of femur fracture: A single center experience. Int J Crit Illn Inj Sci. 2016;6:143-147.

I assume that the “ANOVA test” (Line 123) referred to one-way ANOVA. Were post-hoc comparisons also conducted?

Reply: thanks, corrected. No post-hoc comparisons.

For participant selection, 135 cases (18%) were excluded (Figure 1). Would this lead to over-estimate or under-estimate of the results? Please comment on this in the limitations.

Reply: agree and sentence added to limitations

The “fall from height” and “fall of heavy” were not clearly defined in the methods. Please provide the definitions and examples.

Reply: we added 2 new references and notes for the definition of fall

Line 188, “injuries related to fall from height and fall of heavy object were significantly higher in patients aged 31-59 years”. Since the age range is relatively wide from the young to mid-age. I wonder if those injuries are related to occupations such as construction work.

Reply: agree and reflected in the discussion

Elderly people (≥60 years) usually fall more often than the younger groups, and may suffer hip fracture (femur head or neck). I wonder if there are similar findings from this study. Meanwhile, I wonder why only 37 older adults were included in this study.

Reply: As we can see in table 5, only 3 old patients (>60 yrs) had pelvic injury as well. Furthermore, Table 1 shows that the mean age of the study cohort is low (30.7±16.2).

The authors did a good job in the Discussion. However, in the limitations, several issues may need to be mentioned: (1) representative of the results as they are from one centre (hospital); (2) reliability of the data collection (e.g., injury mechanism); and (3) lack of post-treatment outcomes, which are important to guide the injury management.

Reply: we addressed many points in the limitations. The Hamad trauma center is the only tertiary level 1 center in the country; therefore the results could be representative of the surgically treated femur fracture. The reliability of data was addressed in the method section.  We added the lack of post-treatment outcome in the limitation.

Round 2

Reviewer 2 Report

Why are the following age groups separated in Table 5: <13, 14-30, 31-59,> 60. What is the rationale? Do you have age data before grouping? If so, calculate the mean age for each parameter and use the T-test or the Mann-Whitney test.
Similarly, in table 3, please calculate the average for the time to surgery and compare the significance instead of creating groups 0-6, etc.

Author Response

Why are the following age groups separated in Table 5: <13, 14-30, 31-59,> 60. What is the rationale? Do you have age data before grouping? If so, calculate the mean age for each parameter and use the T-test or the Mann-Whitney test.
Similarly, in table 3, please calculate the average for the time to surgery and compare the significance instead of creating groups 0-6, etc.

Reply:  we classified the age groups to address the burden of femur fracture in each age group as this will help to set the appropriate injury prevention, and management in each group. we selected the age of 13 and below to address this condition on the pediatric group as in our hospital this is age limit to managing trauma patients as pediatric. then the other groups to address the working active age group, adult and old age. we added new raw in table 5 to highlight the mean age in each group. For table 3 , we mentioned in the methods (statistical part) the references for using these time to treat groups and in table 3 we added the median and interquartile range and compared the 3 groups using student t test.